# How Paradoxical Leadership Promotes Employees' Career Sustainability: Evidence from the Chinese Cross-Border E-Commerce Industry

Qi Li [1], Tachia Chin [1,2,*] and Benjian Peng [3]

1    International College, Dhurakij Pundit University, Bangkok 10210, Thailand; 65130480@dpu.ac.th
2    School of Management, Zhejiang University of Technology, Hangzhou 310023, China
3    School of Business, Honghe University, Mengzi 661199, China; pengbenjian183157@163.com
*    Correspondence: tachiachin@zjut.edu.cn

**Abstract:** The ultra-fast development of digital technologies exerts immense pressure on employees in the cross-border e-commerce (CBEC) industry, especially those who are older and have lower levels of education. These employees may appear resistant to digital technologies, which can harm their careers. Leadership can address negative mentalities and sustain employees' career development. Using job demands–resources (JD–R) theory, this study explored how paradoxical leadership can increase employees' career sustainability by mitigating resistance to digital technology. Additionally, the moderating effects of age and education were examined. We analyzed time-lagged data from 141 employees in the Chinese CBEC industry, employing Partial Least Squares–Structural Equation Modeling (PLS–SEM) to test the hypothesized model. Our results revealed that: (1) paradoxical leadership relates negatively to resistance to digital technology but positively to career sustainability; (2) resistance to digital technology negatively relates to career sustainability; (3) resistance to digital technology partially mediates the relationship between paradoxical leadership and career sustainability; (4) age positively moderates the negative relationship between resistance to digital technology and career sustainability, while education negatively moderates that relationship. We offer novel insights into the trade-off effects among the five variables. Furthermore, our study provides practical guidance for firms, emphasizing the critical influence of paradoxical leadership and individual characteristics on ensuring employees' career sustainability in the Chinese CBEC context.

**Keywords:** paradoxical leadership; resistance to digital technology; career sustainability; cross-border e-commerce; China

## 1. Introduction

The rapid development of technology has contributed to the emergence and development of a new business mode, namely, cross-border e-commerce (CBEC). CBEC relies heavily on the development of e-payment platforms, databases, and other digital technologies [1], which has increased the work demands on employees in the CBEC industry by requiring them to master these fast-evolving digital technologies. However, most of the employees in this industry are low-skilled gig workers who lack the resources and time to update and renew their skills. Employers' high requirements therefore place considerable pressure on employees, and this pressure, combined with the fear of being replaced by technology, inevitably causes employees to appear resistant to digital technologies, which is harmful to those wishing to pursue a sustainable career [2,3].

As an essential organizational factor, leadership is believed to ensure employees' long-term career development and mitigate their negative attitudes and actions toward work [4]. However, reducing employees' resistance to digital technology and trying to guarantee their career sustainability in the complex and uncertain CBEC context places high demands on leaders' managerial approaches, as normal managerial skills may not

be sufficient. In addition, changes in the CBEC industry triggered by the implementation of digital technologies inherently include knowledge generation and integration, and require flexibility and discipline, which are polar and paradoxical in nature [5,6]. Therefore, paradoxical leadership, which inherently contains contradictions and paradoxes and was created to solve conflicts brought about by environmental changes, is especially suitable for and may be highly effective in contexts that have high levels of complexity and uncertainty, such as the CBEC industry [7,8]. However, scant research exists on how paradoxical leadership can affect employees' resistance to technology and their career sustainability in the complex CBEC context.

Despite claims that leadership can play a critical role in addressing issues like employees' resistance to change, individual characteristics, such as age, education, and gender, are also believed to be essential factors that may affect employees' adoption of and resistance to technology and the changes that technology brings [9]. An increasing number of researchers are therefore calling for the incorporation of more moderators to further extend the research on technology acceptance [10]. However, few studies have been conducted that consider age, education, or other individual factors as moderators, despite their possible impact on employees' acceptance of and resistance to technology.

In summary, the existing research gaps revolve around exploring the role of paradoxical leadership and individual factors (e.g., age and education) as moderators in the relationship between employees' resistance to digital technology and career sustainability in the Chinese CBEC context. By using JD–R theory, the current study empirically fills these research gaps by combining both organizational and individual factors, namely paradoxical leadership (deemed as job resource), age, and education, to investigate the extent to which these variables can affect employees' resistance to digital technology (deemed as job demand) and their career sustainability (deemed as work-related outcome) in the Chinese CBEC context. Our main contributions are as follows: (1) our findings indicate that paradoxical leadership can address employees' negative mentalities and help them to gain sustainable careers in the CBEC context, which expands the current limited research on what kind of leadership is the most effective in addressing negative issues in an uncertain and complex context. (2) We provide valuable first-hand evidence about the positive relationship between paradoxical leadership and career sustainability, and the mediating role of resistance to digital technology on this relationship. Our research thus provides practical and feasible implications for practitioners and policymakers. It can support them in making wiser decisions to decrease employees' negative outcomes and sustain their jobs in an unpredictable and intricate context. (3) By explaining the moderating role of age and education in the relationship between resistance to digital technology and career sustainability, we further extend the research scope to an individual level and thus enrich the demographic diversity of relevant research, which is beneficial for leaders who aim to tailor policies according to individuals' disparities. (4) Based on JD–R theory, we determine the mechanisms acting between paradoxical leadership, resistance to digital technology, and career sustainability, thereby expanding the scope and applicability of JD–R theory.

## 2. Literature Review and Hypothesis Development

### 2.1. The CBEC Context

With the rapid development of digital technologies, Chinese cross-border e-commerce (CBEC) has emerged as a prominent and transformative business model in the global digital economy. The CBEC industry is a knowledge-intensive ecosystem with interconnected functional subsystems involving a wide array of cross-boundary activities, such as logistics and shipping, electronic payment, customs procedures, and tariff and tax compliance [11]. In addition, the CBEC industry operates in a cross-cultural environment, which introduces a set of challenges marked by high levels of uncertainty and complexity. Such uniqueness makes it into a one-of-a-kind corporate ecosystem and thus different from traditional business models [12].

In this context, the dynamic and uncertain nature of the CBEC industry, which involves reconfigured organizational structures and interactions with stakeholders from diverse cultural backgrounds, may result in career instability for its employees. To address this challenge, an in-depth understanding of a comprehensive management strategy that can help employees to thrive in this unstable and fast-changing environment should be considered. This is crucial for promoting employees' career sustainability and the sustained development of the Chinese CBEC industry.

### 2.2. JD–R Theory

The Job Demands–Resources (JD–R) theory [13] has been a widely recognized framework in the fields of organizational management and human resource management. The theory posits that job characteristics can be divided into two categories: job demands and job resources [13,14].

Job demands refers to the physical, psychological, social, or organizational aspects of a job that require sustained effort and are associated with potential costs for employees [13]. These demands include factors such as high workload and pressure, role ambiguity, and negative mentality. On the other hand, job resources refers to the physical, psychological, social, or organizational aspects of a job that can help individuals to achieve their work goals, reduce job demands, and stimulate personal growth [15]. Job resources include factors like autonomy, social support, resources, and opportunities for development and advancement.

The JD–R theory emphasizes the outcomes of the dynamic interaction between job demands and resources. When job demands exceed resources, it can lead to negative work-related outcomes, such as burnout, work pressure, or reduced job satisfaction, while job resources, associated with increased motivation and work engagement, can buffer the negative outcomes caused by job demands. In addition, the interrelationship between job demands and resources can be contingent on factors such as the occupational sector, level of education, age, and other individual factors [16]. Therefore, given the potential effects of these factors, it is necessary to delve deeper into understanding how specific individual characteristics and contextual factors shape the job demands–resources relationship.

### 2.3. Resistance to Digital Technology and Career Sustainability

The term "resistance to digital technology" has not been specifically defined, but according to Alohali et al. [17], user resistance is "the behavioral expression of a user's opposition to change(s) associated with information system implementation." We therefore inferred that the definition of resistance to digital technology is equivalent. The most accepted definition of career sustainability is "the sequences of career experiences reflected through a variety of patterns of continuity over time" [18]. Based on this definition, Chin et al. [19] proposed a dynamic framework of career sustainability from an interactional perspective and included four intricately interconnected dimensions (i.e., resourceful, flexible, renewable, and integrative) to better measure and understand the connotations of career sustainability.

CBEC is a technology-intensive industry, and technology today is developed so quickly that employees need to constantly learn new skills to keep track of the ever-changing CBEC context. However, most of the employees in this industry are low-paid and short-contracted gig workers who lack sufficient resources and training opportunities to renew their knowledge and skills. According to JD–R theory, when job demands surpass job resources, it can place pressure on employees and negatively affect their work-related outcomes [20]. For instance, such pressure may lead to employees' technological exclusion and resistance, and subsequently cause the appearance of burnout or the intention to quit, which is detrimental to their perceived career sustainability [21,22].

Moreover, Lee et al. [23] found that the adjustments or systematic shocks within an organization will inevitably affect employees' career longevity. Among all the systematic shocks, organizational changes such as changes in technology, personnel, and development goals can place invisible pressure on and cause panic among employees. New technologies

will therefore inevitably lead to resistance among workers, as they may feel overwhelmed, threatened, or disengaged by the high demands and uncertainties associated with the implementation of these new technologies. Such resistance can in turn undermine employees' long-term career prospects. In summary, technological innovations and periods of uncertainty tend to pose significant challenges to individuals and their careers. Following this logic, we hypothesize:

**Hypothesis 1 (H1).** *Resistance to digital technology is negatively related to career sustainability.*

*2.4. Paradoxical Leadership, Resistance to Digital Technology, and Career Sustainability*

Previous studies have indicated that employees' resistance to change and their career sustainability are both affected by social and organizational factors [24,25], and leadership, as an important organizational factor, is considered useful in dealing with both. For instance, Lundy et al. [26] found that in the Canadian public service sector, "project leadership" can effectively reduce employees' resistance to change through proper training. Similarly, Mousa et al. [27] conducted empirical research in Egyptian academia and found that "authentic leadership" can mitigate individuals' resistance to change by fostering organizational learning. In addition, Azanza et al. [28] suggested that "authentic leadership" can help to retain talent by enhancing employees' work engagement in Spain.

However, the effectiveness of these traditional leadership styles has only been confirmed in stable environments, and they seem to be less effective in contexts full of uncertainties and contradictions such as CBEC. Thus, the complex CBEC context demands a new kind of leadership that is more genuine and different from traditional types. In this context, paradoxical leadership is considered more effective because the environment changes more quickly, and therefore it is especially suitable for the unpredictable CBEC conditions [7]. Paradoxical leadership refers to leaders' choice of taking the logic of "both/and" rather than that of "either/or" to tackle the conflicts and challenges brought about by changes in the external environments of enterprises [29]. Zhang et al. [30] combined the Chinese Yin–Yang philosophy and Western paradox theory to define paradoxical leadership as a leaders' "seemingly competing, yet interrelated, behaviors to meet structural and follower demands simultaneously and over time." In addition, paradoxical leadership comprises five managerial approaches: (1) maintaining distance and intimacy; (2) combining egocentrism and other-centrism; (3) allowing autonomy while holding decision-making control; (4) implementing job requirements while endowing flexibility; and (5) treating subordinates equally in a personalized way. Paradoxical leaders use these five methods to cope with all the paradoxes embedded in different contexts.

Maintaining distance and intimacy can help to build a harmonious atmosphere. In a pleasant atmosphere, employees are more willing to stay in their present job to pursue long-term career development. In addition, communication among employees and the sharing of knowledge and resources will be promoted, thereby enabling easier learning and acceptance of digital technologies. Holding decision-making control while allowing autonomy and implementing job requirements while endowing flexibility can provide employees with more rights and flexibility and incentivize them to actively participate in problem-solving. They will then be more likely to embrace new technologies, as these will be seen as new problem-solving methods [31,32]. By treating subordinates equally in a personalized way, leaders can assign tasks according to their employees' abilities and interests. This personal emphasis on the employees will bring them a sense of belonging and identity, which will promote their perceived career sustainability and additional reward behaviors [16,33], such as innovative work behaviors. Employees will then be more receptive to technology rather than both mentally and physically resistant.

In summary, based on the foregoing analysis and JD–R theory, paradoxical leadership can act as a job resource that fosters innovation and encourages employees to embrace technological changes, which consequently helps them to manage and cope with the demands of resistance to digital technology. Additionally, by creating a positive work

environment, paradoxical leadership can lead to increased job satisfaction, engagement, and motivation, which can ultimately result in positive work-related outcomes and improved career sustainability. We therefore hypothesize:

**Hypothesis 2 (H2).** *Paradoxical leadership is negatively related to resistance to digital technology.*

**Hypothesis 3 (H3).** *Paradoxical leadership is positively related to career sustainability.*

*2.5. Mediating Role of Resistance to Digital Technology*

The fast reinvention of technology increases the complexity and uncertainty of jobs, which places considerable pressure on employees. As a result, the high job demands of digital technology inevitably lead to frustration among employees and even resistance to technology. This negative mentality may in turn be detrimental to employees' long-term career development [34,35]. As a leadership style with high levels of flexibility, dynamism, and dialectics, paradoxical leadership can transform negative factors into positive factors, provide sufficient support for employees, and meet their various needs, all of which are conducive to facilitating employees' learning and enhancing their creativity [36]. Accordingly, when managed in a paradoxical way, employees are more likely to accept a new technology and, as a result, their willingness to leave their present work in the aftermath of resistance is less likely to occur. Leaders who practice a paradoxical method of management can thus ensure employees' career sustainability by lowering their resistance to technology first.

This mediating mechanism is reflected in the JD–R model, as high levels of job demands (resistance to digital technology) can negatively impact work-related outcomes (employees' career sustainability), while job resources (paradoxical leadership) can exert positive effects on work-related outcomes (career sustainability). Moreover, job resources can buffer the negative effects of job demands on work-related outcomes [37]. Accordingly, in applying the lens of JD–R theory, we further propose the following mediating relationship:

**Hypothesis 4 (H4).** *Resistance to digital technology mediates the relationship between paradoxical leadership and career sustainability.*

*2.6. Moderating Role of Age and Education*

Research has suggested that the ability to cope with stress is influenced by individual variables [38], but few studies have taken into account its possible effects. In the current study, we therefore considered age and education as two important individual characteristics to test as, at high levels, they can enhance or weaken the negative link between resistance to technology and career sustainability. As individuals become older, their scope of attention tends to become narrow, and their ability to accept and learn new things deteriorates [39]. Consequently, they are more prone to developing a mentality of resistance toward technology, which further strengthens the negative connection between resistance to technology and career sustainability. Conversely, the link between resistance to technology and career sustainability is less likely to be strengthened when individuals are younger or highly educated. The younger generation and highly educated individuals tend to be more open-minded toward newly developed technologies, and their learning abilities are more sophisticated than those of older people [40]. Therefore, the link will be weakened as the level of age decreases and that of education increases. As a result, younger individuals, and those with a high level of education, will be more likely to embrace new technologies rather than reject them, as it is easier for them to keep up with fast-upgrading technologies. Their perceived career sustainability will thus be assured. We therefore inferred that the negative link between resistance to digital technology and career sustainability will be weakened if individuals are young or have a high education level. Hence, we hypothesize:

**Hypothesis 5 (H5).** *Age positively moderates the negative relationship between resistance to digital technology and career sustainability.*

**Hypothesis 6 (H6).** *Education negatively moderates the negative relationship between resistance to digital technology and career sustainability.*

Based on the discussion above, we constructed the research framework shown in Figure 1.

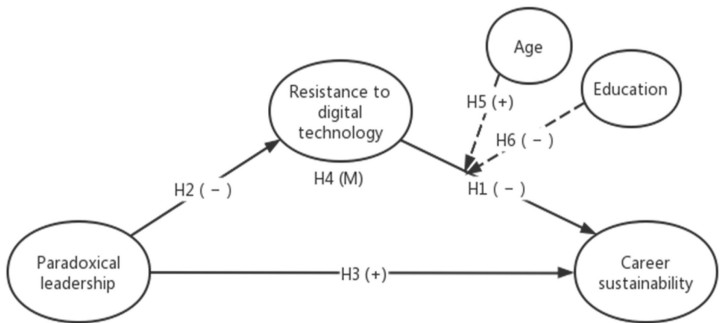

**Figure 1.** The research framework.

## 3. Data and Methodology

### 3.1. Sample and Data Collection

The present study was conducted to investigate the relationship between paradoxical leadership, resistance to digital technology, and career sustainability among Chinese CBEC practitioners. The participants had been engaged in the CBEC industry for more than one year, and most came from the Yunnan, Zhejiang, Guangdong, and Fujian provinces. The data were collected in two stages. The first stage involved an online survey conducted using WJX, a professional online survey platform in China. The second stage involved the offline distribution of questionnaires to the participants. Paradoxical leadership, age, and education were measured during stage 1, while resistance to digital technology and career sustainability were measured a month later for those who responded during stage 1. Before formally conducting our research, we ensured that the participants had a clear understanding of our research purpose, methods, expected outcomes, and potential risks, and we obtained their explicit informed consent to participate in this study. Additionally, all the data collected were handled in accordance with the relevant data protection laws and regulations to protect the participants' privacy.

A total of 208 participants responded at stage 1, and 141 participants responded at stage 2. The final analysis was based on the data obtained from the 141 participants who responded during both stages. The demographic characteristics of the respondents were as follows: 59.3% of the participants were male and 40.7% were female; 78.2% were younger than 40 years of age and 21.8% were older than 40 years old; 71.1% had an undergraduate or lower educational qualification, while 28.9% held a master's degree or higher; 68.6% had worked for less than 20 years and 31.4% had worked for more than 20 years.

### 3.2. Measures

To ensure that the respondents could fully understand all of the questions, which were originally in English, a back-translation procedure was implemented to translate them into Chinese. In this study, we used paradoxical leadership as an independent variable, career sustainability as the dependent variable, resistance to digital technology as the mediating variable, and age and education as moderators.

The independent variable, paradoxical leadership, was measured using the scale developed by Zhang et al. [30] All the items of paradoxical leadership were measured using a six-point Likert scale (1 = strongly disagree to 6 = strongly agree). Sample items were "Treating subordinates uniformly while allowing individualization", "Combining self-centeredness with other-centeredness", and "Maintaining decision control while allowing autonomy" (Cronbach's $\alpha = 0.893$).

The mediating variable, resistance to digital technology, was measured with the scale developed by Nov and Ye [41]. All the items of resistance to digital technology were measured using a six-point Likert scale (1 = strongly disagree to 6 = strongly agree). Sample items included "I generally prefer to use digital technologies that I am familiar with compared to starting to use a new program", "I find it exciting to try out new digital technologies", and "I often feel a bit uncomfortable when trying out new digital technologies, even though they may be beneficial to me" (Cronbach's $\alpha$ = 0.847).

The dependent variable, career sustainability, was measured with the scale developed by Chin et al. [19] All the items under career sustainability were measured using a six-point Likert scale (1 = strongly disagree to 6 = strongly agree). Sample items included "My career makes me feel like I have a bright future", "My career gives me a lot of flexibility", and "My career enables me to integrate information obtained from different sources" (Cronbach's $\alpha$ = 0.947).

### 3.3. Common Method Variance

The occurrence of common method variance can be attributed to the measurement method rather than the underlying construct being measured, leading to possible measurement errors [42–44]. To mitigate this issue, two strategies were implemented in the present study. First, in terms of the questionnaires, the use of a paginated scale and allowing the participants sufficient time to answer each page minimized the effect of common method variance caused by the same continuity scale [45]. Second, Harman's single-factor test was conducted to determine the presence of common method variance. Based on the exploratory factor analysis, the first factor explained only 33.176% of the variance, which was lower than the 50% threshold. There was therefore no significant common method variance.

## 4. Results

### 4.1. Measurement Model

In this study, we used the SmartPLS 3.3.9 package to evaluate the hypotheses in our research model [45]. We employed the measurement model to assess the reliability, convergent validity, and discriminant validity of our data [46,47]. To analyze the intricate relationships between latent and observed variables, we used Partial Least Squares–Structural Equation Modeling (PLS–SEM) for both the measurement and structural model analysis.

PLS–SEM is a widely accepted variance-based, descriptive, and prediction-oriented approach to structural equation modeling [48]. It is a statistical approach to modeling complex multivariable relationships among observed and latent variables, and it combines principal component analyses, path analyses, and regression to evaluate theory and data simultaneously. PLS–SEM has been widely adopted for analysis in research on tourism management, organizational management, human resources management, and marketing management [49,50].

PLS–SEM method uses a two-step approach. The first step requires the assessment of the measurement model (outer model), which allows the relationship between the observable variables and theoretical concepts to be specified. In the second step, the structural model (inner model) must be evaluated to test the extent to which the relationship specified by the proposed model is consistent with the acquired data [51].

Using PLS–SEM allows for greater flexibility in handling various complex modeling problems where it is difficult or impossible to meet the hard assumptions of other traditional multivariate statistics. Therefore, it is especially suitable for our study, which has relatively small samples (n = 141) and a complex research model (combing mediating and moderating effects).

For reliability, internal consistency was ensured by determining the composite reliability of the constructs [52]. The Cronbach's $\alpha$ values for each dimension ranged from 0.847 to 0.947 (resistance to digital technology [RDT] and career sustainability [CS], respectively) and were thus higher than the recommended value of 0.7. The combined reliability ranged from 0.881 to 0.954 (RDT and CS, respectively), which were all higher than 0.8. In addition, the

consistent PLS method was used to correct the estimate of the measured structure with a new reliability coefficient, rho_A, which ranged from 0.877 to 0.952 (RDT and CS, respectively). These results confirmed the high internal consistency of the measurements [53–55].

In terms of convergent validity, the factor loadings of all the items were significant (>0.7). The average variance extracted (AVE) values ranged from 0.557 to 0.703 (RDT and paradoxical leadership [PL], respectively) and were thus higher than 0.5. The convergent validity of these measures was therefore satisfactory.

The discriminant validity of the constructs was evaluated using the approach described by Fornell and Larcker [52]. In this way, all the square roots of the AVE values were higher than all the correlation coefficients, which indicated a satisfactory discriminant validity of the measures [52,56]. All the data presented in Tables 1 and 2 satisfied the aforementioned criteria, thus demonstrating the high reliability, convergent validity, and discriminant validity of the data.

**Table 1.** Confirmatory factor analysis results for the measured variables.

| Construct | Items | Factor Loading | A | rho_A | CR | AVE | VIF |
|---|---|---|---|---|---|---|---|
| Paradoxical Leadership (PL) | PL1 | 0.869 | | | | | |
| | PL2 | 0.873 | | | | | |
| | PL3 | 0.862 | 0.893 | 0.895 | 0.922 | 0.703 | 1.000 |
| | PL4 | 0.859 | | | | | |
| | PL5 | 0.721 | | | | | |
| Resistance to Digital Technology (RDT) | RDT1 | 0.728 | | | | | |
| | RDT2 | 0.701 | | | | | |
| | RDT3 | 0.870 | 0.847 | 0.877 | 0.881 | 0.557 | 1.375 |
| | RDT4 | 0.859 | | | | | |
| | RDT5 | 0.866 | | | | | |
| | RDT6 | 0.753 | | | | | |
| Career Sustainability (CS) | CS1 | 0.719 | | | | | |
| | CS2 | 0.804 | | | | | |
| | CS3 | 0.844 | | | | | |
| | CS4 | 0.843 | | | | | |
| | CS5 | 0.847 | | | | | |
| | CS6 | 0.815 | | | | | |
| | CS7 | 0.824 | 0.947 | 0.952 | 0.954 | 0.635 | DV |
| | CS8 | 0.811 | | | | | |
| | CS9 | 0.807 | | | | | |
| | CS10 | 0.804 | | | | | |
| | CS11 | 0.819 | | | | | |
| | CS12 | 0.809 | | | | | |

Note: PL: paradoxical leadership; RDT: resistance to digital technology; CS: career sustainability.

**Table 2.** Discriminant validity analysis (Fornell and Larcker).

| Construct | PL | RDT | CS |
|---|---|---|---|
| PL | 0.839 | | |
| RDT | −0.522 | 0.746 | |
| CS | 0.984 | −0.540 | 0.797 |

Note: PL: paradoxical leadership; RDT: resistance to digital technology; CS: career sustainability.

Introduced by Henseler et al. [57], the Standardized Root Mean Squared Residual (SRMR) is a statistic used to assess the model fit in structural equation modeling (SEM) and Partial Least Squares–Structural Equation Modeling (PLS–SEM). It measures the discrepancy between the observed covariance matrix and the model-implied covariance matrix. The SRMR is a standardized version of the root mean squared residual, which allows for comparison across different models and datasets.

The SRMR value ranges from zero to one, with lower values indicating a better model fit. A common threshold for a well-fitted model is an SRMR value less than 0.10. If

the SRMR is close to or below 0.10, it suggests that the model is a good representation of the observed data. On the other hand, an SRMR value above 0.10 may indicate a potential model misfit, and further investigation or model refinement may be necessary. The SRMR in the current study was 0.098, thus indicating a favorable model. With regard to multicollinearity, according to Hair et al. [21], value tolerance should have a variance inflation factor (VIF) value below five. Table 2 presents all of the construct VIF values ranging from 1.000 to 1.375, which indicates that the results met the requirements.

### 4.2. Structural Model

To test the hypotheses, the bootstrap resampling method in SmartPLS was used to evaluate the PLS results, and the responses were resampled 5000 times [58]. Table 3 presents the results. The overall $R^2$ value was 0.256, and the results supported hypotheses H1, H2, H3, and H4. The results suggested that resistance to digital technology is negatively related to career sustainability (H1, $\beta = -0.036$, $p < 0.05$); paradoxical leadership is negatively related to resistance to digital technology (H2, $\beta = -0.522$, $p < 0.01$); paradoxical leadership is positively related to career sustainability (H3, $\beta = 0.984$, $p < 0.01$); and resistance to digital technology mediates the relationship between paradoxical leadership and career sustainability (H4, $\beta = 0.019$, $p < 0.05$).

**Table 3.** Hypothesis constructs.

| Hypothesis | Effect | T-Value | *p*-Value | Result |
|:---:|:---:|:---:|:---:|:---:|
| H1: RDT → CS | −0.036 | 2.498 | 0.013 | Significant |
| H2: PL → RDT | −0.522 | 6.702 | 0.001 | Significant |
| H3: PL → CS | 0.984 | 9.786 | 0.001 | Significant |
| H4: PL → RDT → CS | 0.019 | 2.111 | 0.035 | Significant |
| H5: age × RDT → CS | 0.027 | 3.750 | 0.001 | Significant |
| H6: education × RDT → CS | −0.036 | 4.001 | 0.001 | Significant |

Note: paradoxical leadership (PL), resistance to digital technology (RDT), career sustainability (CS).

Furthermore, our study results confirmed the moderating effects of age and education on the relationship between resistance to digital technology and career sustainability. The findings demonstrated the positive moderating effect of age (H5, $\beta = 0.027$, $p < 0.01$) and the negative moderating effect of education (H6, $\beta = -0.036$, $p < 0.01$). Figure 2 presents PLS results of the research model.

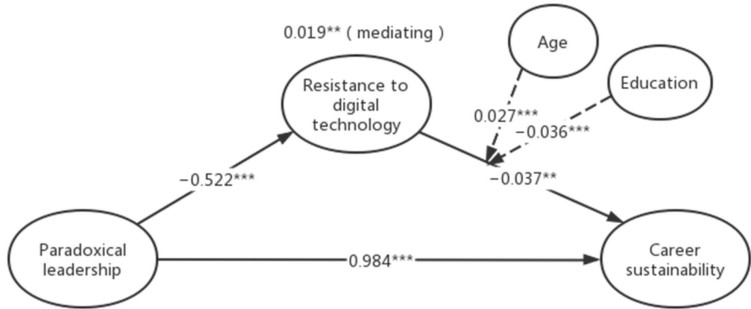

**Figure 2.** Path diagram and the standardized estimates of the model. (Note: *** means that the correlation is significant at the level of 0.001; ** means that the correlation is significant at the level of 0.05).

To better illustrate the moderating effects of age and education, interaction effects were plotted. As shown in Figure 3, the negative relationship between resistance to digital technology and career sustainability was more distinct for employees of an older age and with a lower education level than for those of a younger age with a higher education level.

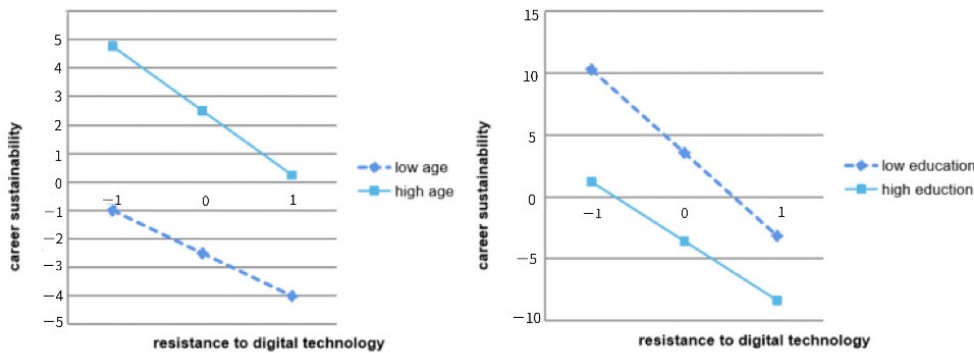

**Figure 3.** The moderating effects of age and education.

All of the hypotheses were thus verified, and the partial mediating effect of resistance to digital technology on the relationship between paradoxical leadership and career sustainability was evident.

*4.3. Robustness Tests*

4.3.1. Endogeneity

Following a suggestion by Hult et al. [59], this study employed Park and Gupta's [60] Gaussian copula approach to assess the potential endogeneity of the proposed model. We first examined whether the variables with potential endogeneity (i.e., paradoxical leadership, resistance to digital technology, age, and education) were non-normally distributed by running the Kolmogorov–Smirnov test with Lilliefors correction [61]. The results showed that none of the variables had normally distributed scores, which met the requirement for conducting a Gaussian copula analysis [59].

The results in Table 4 show non-significant Gaussian copulas of $-0.052$ for PL ($p$ value = 0.28), 0.076 for RDT ($p$ value = 0.598), $-0.023$ for AGE ($p$ value = 0.904), and $-0.082$ for EDU ($p$ value = 0.673). Furthermore, we have checked all the combinations of Gaussian copulas included in the model and none is significant (see Table 4). We consequently conclude that endogeneity is not present in this study, which supports the robustness of the structural model results in this regard [59].

**Table 4.** Assessment of endogeneity test using the Gaussian copula approach.

| Test | Coefficient | $p$ Value |
| --- | --- | --- |
| Gaussian copula of model 1 (endogenous variable: PL) | $-0.052$ | 0.280 |
| Gaussian copula of model 2 (endogenous variable: RDT) | 0.076 | 0.598 |
| Gaussian copula of model 3 (endogenous variable: AGE) | $-0.023$ | 0.904 |
| Gaussian copula of model 4 (endogenous variable: EDU) | $-0.082$ | 0.673 |
| Gaussian copula of model 5 (endogenous variable: PL, RDT) | $-0.058$ | 0.217 |
| | $-0.189$ | 0.716 |
| Gaussian copula of model 6 (endogenous variable: PL, AGE) | $-0.051$ | 0.295 |
| | $-0.031$ | 0.872 |
| Gaussian copula of model 7 (endogenous variable: PL, EDU) | $-0.058$ | 0.256 |
| | $-0.131$ | 0.464 |
| Gaussian copula of model 9 (endogenous variable: RDT, EDU) | $-0.167$ | 0.170 |
| | $-0.017$ | 0.931 |
| Gaussian copula of model 10 (endogenous variable: AGE, EDU) | $-0.085$ | 0.666 |
| | $-0.033$ | 0.865 |
| Gaussian copula of model 11 (endogenous variable: PL, RDT, AGE) | $-0.060$ | 0.231 |
| | $-0.204$ | 0.083 |
| | $-0.108$ | 0.529 |
| Gaussian copula of model 12 (endogenous variable: PL, AGE, EDU) | $-0.058$ | 0.250 |
| | $-0.047$ | 0.800 |
| | $-0.135$ | 0.458 |

**Table 4.** *Cont.*

| Test | Coefficient | *p* Value |
|---|---|---|
| Gaussian copula of model 13 (endogenous variable: RDT, AGE, EDU) | −0.092 | 0.614 |
| | −0.021 | 0.919 |
| | −0.179 | 0.161 |
| Gaussian copula of model 14 (endogenous variable: PL, RDT, AGE, EDU) | −0.062 | 0.219 |
| | −0.192 | 0.118 |
| | −0.112 | 0.515 |
| | −0.070 | 0.710 |

Note: PL: paradoxical leadership; RDT: resistance to digital technology; AGE: age; EDU: education.

### 4.3.2. Heterogeneity

To identify unobserved heterogeneity in PLS path models, we carried out the FIMIX–PLS procedure [62]. Following Matthews et al. [63], we initiated the procedure with a maximum number of iterations of 5000 and 10 repetitions. Taking into account the number of the variables in our research model and following the instructions of Hair et al. [64], we finally got the minimum sample size of our study (i.e., 25). Dividing the sample size (i.e., 141) by the minimum sample size (i.e., 25) yields an upper bound of 5.64. Given the complexity of the model and ensuring compliance with the minimum sample size requirement, we choose a maximum of five segments rather than six [63]. Subsequently, FIMIX–PLS was run for one to five segments.

The results of the one to five segments (Table 5) paint an ambiguous picture: (1) AIC3 and CAIC point to different segments, AIC3 pointing to segment five while the AIC5 points to segment three; (2) AIC4 and BIC both indicate segment four, which is different from segment three or five; (3) according to AIC and MDL5, the appropriate segment should lower than five and higher than one, but the EN value points to five, which overestimates the correct number. Therefore, the number of segments is ambiguous, indicating that there is no unobserved heterogeneity problem in current research.

**Table 5.** Fit indices for the one to five segment solutions.

| Criteria | Number of Segments | | | | |
|---|---|---|---|---|---|
| | 1 | 2 | 3 | 4 | 5 |
| AIC | 392.06 | 348.053 | 311.136 | 287.079 | **278.896** |
| AIC3 | 399.06 | 363.053 | 334.136 | 318.079 | **317.896** |
| AIC4 | 406.06 | 378.053 | 357.136 | **349.079** | 356.896 |
| BIC | 412.651 | 392.178 | 378.793 | **378.27** | 393.62 |
| CAIC | 419.651 | 407.178 | **401.793** | 409.27 | 432.62 |
| HQ | 400.428 | 365.984 | 338.63 | **324.136** | 325.517 |
| MDL5 | **551.017** | 688.676 | 833.424 | 991.033 | 1164.516 |
| LnL | −189.03 | −159.026 | −132.568 | −112.539 | −100.448 |
| EN | NA | 0.7 | 0.635 | 0.731 | **0.765** |
| NFI | NA | 0.713 | 0.597 | 0.689 | 0.707 |
| NEC | NA | 42.009 | 51.093 | 37.594 | 32.943 |

Note: AIC: Akaike's information criterion. AIC3: modified AIC with factor three; AIC4: modified AIC with factor four; BIC: Bayesian information criteria; CAIC: consistent AIC; HQ: Hannan Quinn criterion; MDL5: minimum description length with factor five; LnL: log likelihood; EN: entropy statistic; NFI: non-fuzzy index; NEC: normalized entropy criterion; NA: not available. Numbers in bold indicate the best outcome per segment retention criterion.

### 4.3.3. Nonlinear Effects

To test for potential nonlinear effects in the structural model relationships, we included interaction terms to represent the quadratic effects of paradoxical leadership on resistance to digital technology, and those of paradoxical leadership and resistance to digital technology on career sustainability. The results of bootstrapping with 5000 samples show that none of

the nonlinear effects are significant (Table 6). We therefore conclude that the linear effects model is robust.

**Table 6.** Assessment of nonlinear effects.

| Nonlinear Relationship | Coefficient | Percentile Confidence Interval | t Statistic | *p* Value | f$^2$ |
|---|---|---|---|---|---|
| PL × PL → RDT | 0.003 | [−0.073, 0.094] | 0.068 | 0.946 | 0.01 |
| PL × PL → CS | −0.021 | [−0.04, 0.001] | 1.960 | 0.051 | 0.06 |
| RDT × RDT → CS | 0.004 | [−0.021, 0.028] | 0.311 | 0.756 | 0.01 |

Note: PL: paradoxical leadership; RDT: resistance to digital technology; CS: career sustainability. Quadratic effects assessed based on a two-tailed percentile bootstrapping test at 5% confidence level [2.5%, 97.5%].

## 5. Discussion

Our study results support the linkages between paradoxical leadership, resistance to digital technology, career sustainability, age, and education. First, our results indicate that in terms of direct relationships, resistance to digital technology is negatively related to career sustainability, and paradoxical leadership is negatively related to resistance to digital technology but positively related to career sustainability, which supports H1, H2, and H3. The study by Srivastava [34] suggested that resistance to change will trigger employees' turnover intentions and thus be a solid obstacle on their path of career development. In addition, Zhang et al. [65] found that paradoxical leadership has positive effects on sustaining employees' careers and enables them to exhibit more proficient, adaptive, and proactive behaviors. Accordingly, the cases above both directly and indirectly support our results. Second, our results suggest that paradoxical leadership can increase employees' career sustainability by mitigating their resistance to digital technology, which confirms the mediating role of resistance to digital technology and supports H4. This is the first study to probe this mediating mechanism between the three variables. Finally, our results demonstrate that the negative relationship between resistance to digital technology and career sustainability may be more pronounced among older and less educated individuals, which supports H5 and H6. The findings of Srivastava [34] suggested that age was significantly associated with negative work outcomes, such as job burnout and intention to quit caused by resistance to change. In addition, Binyamin et al. [66] found that people with higher educational levels showed more willingness and intention to accept digital technologies. Thus, the findings above further support our results.

### 5.1. Implications for Theories

The aforementioned outcomes have three main theoretical implications for future research. First, this study enriches the volume of literature on what kind of leadership can effectively address negative work-related outcomes in a complex context by taking paradoxical leadership into account and confirming that it can reduce employees' resistance to digital technology and ensure their career sustainability in the Chinese CBEC industry. This further strengthens the understanding and applicability of paradoxical leadership and provides evidence to support the claim that paradoxical leadership is especially suitable for complex and uncertain contexts [5]. Second, through the lens of JD–R theory, we inferred from our results that paradoxical leadership (job resources) can improve employees' career sustainability (work-related outcomes) by reducing resistance to digital technology (job demands). This is the first study to probe such a mediating mechanism between the three variables, and it thus contributes to the current body of research concerning JD–R theory, further extending its research scope. Finally, our results show that career sustainability among older people and those with low levels of education is more likely to decrease in the aftermath of resistance to digital technology. This adds to the limited extant research on how individual factors can act on employees' technological resistance. Our findings provide further support for the importance of treating subordinates in a personalized way

based on their individual traits, which contributes to a greater diversity of demographics in this field of research.

### 5.2. Implications for Practice

This study provides several practical implications for managers. First, according to our findings, paradoxical leadership is useful in addressing employees' resistance to digital technology and decreased career sustainability in the aftermath of resistance. Managers should therefore cultivate their contradictory and "both/and" thinking and be encouraged to employ a paradoxical method of management. For instance, they should maintain an inclusive and open attitude to handling work affairs and meeting employees' needs. They should further delegate appropriate decision-making powers to subordinates to improve their subordinates' feelings of respect and recognition from their superiors, allowing them to experience themselves in a flexible and open environment and thereby maintaining a more pleasant mood. This positive emotion may further inspire employees' passion, promote positive attitudes toward technology, activate innovative work behaviors, and stimulate employees' desires to stay in their current positions and seek sustainable career development.

Second, an older age and lower education can deepen the negative relationship between resistance to digital technology and career sustainability. Leaders in human resources management should therefore take into account employees' demographic factors and provide more resources to help renew their abilities and guide their work based on these individual characteristics, so as to enhance the alignment between the organization and its workforce. The aforementioned approaches are expected to help reduce employees' negative work outcomes and ensure their career sustainability in today's fast-changing CBEC context. They may also be helpful for CBEC enterprises seeking to maintain their competitiveness and ongoing development.

### 5.3. Limitations and Future Directions

Although the present study yielded some noteworthy findings, there are still some limitations that point to future research.

#### 5.3.1. Individual Factors

First, we only investigated the moderating effects of age and education. Future studies should take into account more individual factors, such as gender, experiences, and years of work, to test the extent to which these factors may affect the relationship between resistance and career sustainability, and thus add demographic diversity to this research area.

#### 5.3.2. Sample Size

Second, the sample size in our study was small and thus it is challenging to measure the exact link between the different variables with accuracy and rigor. Future research should therefore aim to further extend the validity of the findings through the use of a larger sample to test the proposed model.

#### 5.3.3. Context

Lastly but importantly, we conducted this research in the Chinese CBEC context. The CBEC context may be different in other countries, and therefore similar studies should be conducted in the CBEC context in other countries.

## 6. Conclusions

Our study provides valuable firsthand knowledge about the impact of context-dependent paradoxical leadership on employees' resistance to digital technology and career sustainability in the Chinese CBEC context. The fast digital technology advancements have laid a solid technological foundation for the rapid development of CBEC in the post-pandemic era; however, many low-skilled gig workers in this industry appear to be resistant to their use.

Such negative psychology can threaten the sustainable development of employees' careers and that of the CBEC industry. It is thus of great significance to study in depth the kind of leadership style that may be most useful in the complex and uncertain CBEC industry. Regarding this perspective, we provided evidence that paradoxical leadership can be workable in addressing employees' resistance to digital technologies and decreased career sustainability caused by negative mentalities in the complex CBEC context. In addition, we investigated the deepening and weakening effects of age and education on the link between resistance to digital technology and career sustainability and have thus provided a more nuanced understanding of the effects of demographic factors.

In summary, the scientific contribution of the current study lies in advancing the understanding of how paradoxical leadership increases career sustainability by mitigating resistance to digital technology and revealing the moderating effects of age and education. These implications provide theoretical and practical implications and insights for future research and thereby contribute to scientific readership in the fields of leadership, technology adoption, career sustainability, and JD–R theory.

## 7. Policy Recommendations

The current Chinese CBEC industry still has limitations, such as complex regulatory procedures and a lack of support and guidance on the implementation of new technologies. These restrict the application and innovation of technology to some extent. We therefore propose several policy recommendations:

(1) The provision of digital technology training. The government and enterprises should collaborate to provide digital technology training and development opportunities for employees in the CBEC industry, especially those of an older age and with lower educational backgrounds, as this will enhance their skills and help them to overcome their resistance to digital technology, thus improving their career sustainability.

(2) Pay more attention to vulnerable groups. Measures should be taken by the government and organizations to promote age diversity and inclusiveness, and to avoid age and education discrimination and inequality. Special policy attention should be given to older and less educated employees, and equal opportunities and resources should be provided to ensure their career sustainability and to allow them to benefit from the development of digital technology.

(3) The impact of political factors on technology adoption and career development is an intriguing area for further investigation. We encourage future researchers to explore the influence of political factors and their interactions with other variables to provide a more comprehensive understanding of the dynamics within the Chinese CBEC industry.

**Author Contributions:** Q.L. designed the research and wrote the majority of the manuscript; T.C. provided guidance throughout the entire process and offered final modification suggestions; B.P. collected and analyzed the data. All authors have read and agreed to the published version of the manuscript.

**Funding:** This research was supported by the National Science Foundation of China (Grant No. 72272136).

**Institutional Review Board Statement:** Not applicable.

**Informed Consent Statement:** Informed consent was obtained from all subjects involved in the study.

**Data Availability Statement:** The data that support the findings of this study are available from the corresponding author.

**Acknowledgments:** We thank the editor and reviewers for their valuable comments and suggestions.

**Conflicts of Interest:** The authors declare no conflict of interest.

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
