# Peer review of "How Paradoxical Leadership Promotes Employees’ Career Sustainability: Evidence from the Chinese Cross-Border E-Commerce Industry"

_sustainability, doi:10.3390/su151612407_

Round 1
Reviewer 1 Report
Dear authors!
I enjoyed your research.
However, let me make some remarks.
The article leaves an ambiguous impression
On the one hand, it is written in good language, well-structured, parts of the article are correctly distributed, and the necessary points are spelled out.
On the other hand, some of her statements often look banal, as if the authors are trying to tell everyone long-known facts, referring to various "respectable" studies. In general, we do not deny these statements, but we consider it unnecessary to confirm them with references to studies.
These moments include, for example
• The fact that the older a person is, the more difficult it is for him to accept innovations
• That adults are increasingly resistant to innovation
• That new technologies cause resistance among workers (lines 116-117)
• That in general, innovations and the period of uncertainty cause difficulties for people
Isn't this what the well-known saying “God forbid you live in an era of change” says?
• That younger people as well as more educated people are more accepting of change (208-209)
• That good management can reduce resistance to change (line 130-131), and that through various manipulations of people (you call it "genuine leadership" you can retain talent and increase people's involvement in work
I think that in this aspect the article needs to be reworked a little.
In addition, it is not very clear what is new in the idea that the authors come to, namely, that , since an elder age and lower education can deepen the negative relationship between resistance to digital technology and career sustainability, leaders in the human resources management should take into account employees’ demographic factors, and provide more resources to help renew their abilities and guide their work based on these individual characteristics, so as to enhance the alignment between the organization and its workforce. (411-417)
However, you raise an important issue of our time, namely the ultra-fast development of digital technologies and the fact that it is extremely difficult for the older generation to get used to this new reality. Like everyone else, at all levels of personal and public life, it is difficult to accept this pace of development of new technologies.
Because really the correlation between the speed of development of technologies and, as a result, socio-economic and infrastructural transformations with human life allows us to state a qualitative jump in the speed of development, marking the transition to a new temporal era.
And in these realities, it is necessary to comprehend the foundations of the transformation of the socio-economic infrastructure and the entire "life world", thereby denoting the specifics of the "future that has come" that already exists before our eyes, causing an endless series of changes that have raised a number of important questions in all areas.
Including the problem of leadership in this new world, in which the realities are determined by the very rapid changes caused by the progress of digital technologies.
Good luck with your scientific research
Author Response
Thank you for your valuable feedback. We have tried our best to revise our manuscript according to your professional comments, and the whole revised manuscript has been edited by SCRIBENDI, a translation agency, to ensure the quality of our english.
More detailed revisons in line with your suggestions please see the attachment:

Reviewer 2 Report
Thank you for your submission, this is a well written manuscript which identifies the implications of the research, has a clear analysis using appropriate testing and a supported methodology.
My only suggestions would be to add more supporting literature in the review and to further justify the design and ethical considerations in the methodology.
Thank you for sharing, this was an interesting read which will take research forward in the area.
No issues, the level of English is fine.
Author Response

(The authors gave the same response as above.)

Reviewer 3 Report
Sustainability 2023, 15, x FOR PEER REVIEW 4 of 13
-
It is not clear in with way the leadership five dimensions are related to path of relation analysed in the article. The CFA model is a good solution but definite relation of leaderships are also important.
-
Author Response

(The authors gave the same response as above.)

Reviewer 4 Report
Title:
How paradoxical leadership increases employees’ career sustainability by mitigating their resistance to digital technology
First of all, I would like to thank you for the review invitation. I have reviewed this exciting article. This paper states that intensifying digitalization, reducing the negative effects caused by employees’ resistance to digital technology have also become increasingly important. Given leadership styles are believed to mitigate such resistance, combined with the JD-R theory, study aimed to explore how a specific type of paradoxical leadership increase employees’ career sustainability by mitigating their resistance to digital technology. To test our hypotheses, we conduct both online and offline questionnaire surveys on Chinese employees in the cross-border e-commerce industry and use structural equation modeling to do the analysis. Results show that: (1) paradoxical leadership negatively relates to resistance to digital technology but positively relates to career sustainability; (2) resistance to digital technology negatively relates to career sustainability; (3) paradoxical leadership increases career sustainability through mitigating resistance to digital technology.
Abstract
The abstract quality is not high standard to meet the merit of scientific writing that needs a high standard of writing to publish in outstanding journals like SUSTAINABILITY. Please revise the whole article and remove English grammar problems. I suggest the authors take English editing services from some agencies to improve the quality of this study.
Introduction section
I suggest that authors to read the suggested studies add the latest citations to the introduction, literature and method sections to enhance the quality of the study.
Wang, S., Al-Sulaiti, K., & Shah, S. A. R. (2023, 2023/06/01). The Impact of Economic Corridor and Tourism on Local Community’s Quality of Life under One Belt One Road Context. Evaluation Review, 47(3), 445-454.
Literature section:
Add literature section. You cannot delete this section. Read the suggested studies and cite these papers in the literature to enhance the quality of your work.
Shah, S. A. R., Zhang, Q., Tang, H., & Al-Sulaiti, K. I. (2023, 2023/06/01/). Waste management, quality of life and natural resources utilization matter for renewable electricity generation: The main and moderate role of environmental policy. Utilities Policy, 82, 101584. https://doi.org/10.1016/j.jup.2023.101584
Materials and Methods
This section is very weak. Please follow the suggested studies and improve your paper. The authors need to improve this section. I am recommending some good studies. Read the methods of these studies, improve your paper, and cite these studies in this section. Suggested useful articles citations:
Local Burden of Disease, H. I. V. C. (2021). Mapping subnational HIV mortality in six Latin American countries with incomplete vital registration systems. BMC Medicine, 19(1), 4. doi:10.1186/s12916-020-01876-4
Discussion section:
I suggest you to discuss the current situation of the COVID-19 pandemic. How it has affected tourism activities. Read the proposed studies to improve your analysis. See the recommended studies and improve your sections.
Implications
Explain this section effectively. It needs a better presentation related to the study topic.
Limitations
Discuss the study’s limitations with a separate heading and discuss it briefly.
Policy recommendations
Policy recommendations are not sufficient at this stage of the manuscript. The authors must add a separate section for policy recommendations in the conclusion section. Also, add some exciting limitations regarding political factors for future studies.
Conclusion
The conclusion section needs improvement and the authors need to expand it as it will improve the quality of this study. The English level needs some improvement to reach a satisfactory level, specifically the grammar. It should sufficiently meet quality to reach scientific merit for publication. I recommend that the authors describe the study's scientific contribution to the existing body of knowledge in the discussion section. How does this study’s implications provide useful information for the scientific readership? I endorse this manuscript for publication after minor corrections, as suggested.
Moderate editing of the English language is required.
Author Response

(The authors gave the same response as above.)

Round 2
Reviewer 4 Report
How paradoxical leadership increases employees’ career sustainability by mitigating their resistance to digital technology
I am glad to assess the revised manuscript. Thank you for the opportunity to review the revised paper. It was an insightful and well-written piece that provided a new perspective on the topic under consideration.
It is a good research topic offered in this manuscript. The topic of this research study has a practical significance to the scientific knowledge. The authors have investigated a good research area. I will accept this article after some changes. Modify according to these suggestions.
Before accepting this study for publication, I suggest strong literature support. I suggest the authors to cite all these studies to improve the quality. However, the authors need to revise it according to my minor suggestions. The title needs clarity with the design of the main study. I suggest authors revise their title with a better and more suitable words. See the below-recommended studies to improve your TTLE and Abstract quality. Cite all the suggested studies below to improve the quality. I will accept this paper after these minor changes.
Abstract
First, I have some suggestions for the authors to enhance the quality of this innovative study. Please write a high-quality abstract, as it is the main door of the study. I suggest authors add a Graphical Abstract meaningfully to reflect the whole idea. Remove minor grammar errors.
Introduction section
This section needs improvement. Please read these studies, revise your abstract, and cite them in the introduction and literature part. Cite the suggested studies to improve the quality. The introduction is not well established with the support of the study objectives and fresh literature evidence. The introduction should benefit from executing further improvement in the organization and clarity of the study argument. I invite the author to define the topic's gap and indicate how the paper fills the gap. Cite these studies to strengthen the quality of this study.
Abaalzamat, K. H., Al-Sulaiti, K. I., Alzboun, N. M., & Khawaldah, H. A. (2021). The Role of Katara Cultural Village in Enhancing and Marketing the Image of Qatar: Evidence From TripAdvisor. SAGE Open, 11(2), 21582440211022737. doi:10.1177/21582440211022737
Al-Sulaiti, K. I., Abaalzamat, K. H., Khawaldah, H., & Alzboun, N. (2021). Evaluation of Katara Cultural Village Events And Services: a Visitors' Perspective. Event Management, 25(6), 653-664. doi:10.3727/152599521x16106577965099
Literature review
This study review needs improvement with the latest support of the literature. Furthermore, I have found that the literature review used is not updated. Thus, updating the literature review part with fresh studies is necessary. Ideally, the literature review after the introduction will be added separately. At the end of the introduction part, please indicate a theoretical contribution of the paper and add the structure of the paper by sections. I recommend the author find the literature gap and indicate how the paper fills that gap. Cite these studies to strengthen the quality of this study.
Li, X., Dongling, W., Baig, N. U. A., & Zhang, R. (2022). From Cultural Tourism to Social Entrepreneurship: Role of Social Value Creation for Environmental Sustainability. Front Psychol, 13, 925768. doi:10.3389/fpsyg.2022.925768
Methodology: The section could be more explicit about the research question or hypotheses being tested. Furthermore, the methodology could be more detailed about the statistical methods used to analyze the data. Additionally, the section could benefit from more information about how missing data was handled, what assumptions were made about the distribution of the outcome variables, and how the model fit was assessed. I recommend including the data part inside the methodology and naming the section Data and Methodology. Cite these studies in the methodology to improve the study.
Li, Y., Al-Sulaiti, K., Dongling, W., & Al-Sulaiti, I. (2022). Tax Avoidance Culture and Employees' Behavior Affect Sustainable Business Performance: The Moderating Role of Corporate Social Responsibility. Frontiers in Environmental Science, 10. doi:10.3389/fenvs.2022.964410
Discussion
Improve this discussion. Argue on the results and cite more studies to support the discussion. Cite these studies.
Result: I recommend the author add robustness test results to verify the results' correctness. Furthermore, it is necessary to add a discussion part to interpret the result obtained in detail.
Implications
Explain this section effectively. It needs a better presentation related to the study topic.
Limitations
Discuss the study’s limitations with a separate heading and discuss it briefly.
Policy recommendations
Policy recommendations are not sufficient at this stage of the manuscript. The authors must add a separate section for policy recommendations in the conclusion section. Also, add some exciting limitations regarding political factors for future studies.
Conclusion: The conclusion provides a comprehensive summary of the findings. The conclusion could have been strengthened by acknowledging some limitations of the study, such as potential confounding variables that may have affected the results. The conclusion could have also provided some recommendations for future research. I recommend the author compare the results obtained in the study with the findings of other authors and explain how the results relate to each other.
Moderate editing of the English language is required.
Author Response
We are very grateful for your thoughtful comments and suggestions. Your helpful comments and suggestions greatly improve the quality of this manuscript. According to the comments, we revised this manuscript correspondingly again. Significant changes were made to the paper and all the revised contents were highlighted in red within the newly revised manuscript. Our point-by-point responses to you please see the attachment:

Round 3
Reviewer 4 Report
I have reviewed the revised version of this manuscript. The paper has been revised in a good manner. I endorse this paper for publication in it's current format. Good luck
It is acceptable.